# Mucosal Immune Defence Gene Polymorphisms as Relevant Players in the Pathogenesis of IgA Vasculitis?

**DOI:** 10.3390/ijms241713063

**Published:** 2023-08-22

**Authors:** Joao Carlos Batista-Liz, Vanesa Calvo-Río, María Sebastián Mora-Gil, Belén Sevilla-Pérez, Ana Márquez, María Teresa Leonardo, Ana Peñalba, Francisco David Carmona, Javier Narvaez, Luis Martín-Penagos, Lara Belmar-Vega, Cristina Gómez-Fernández, Luis Caminal-Montero, Paz Collado, Patricia Quiroga-Colina, Miren Uriarte-Ecenarro, Esteban Rubio, Manuel León Luque, Juan María Blanco-Madrigal, Eva Galíndez-Agirregoikoa, Javier Martín, Santos Castañeda, Miguel Angel González-Gay, Ricardo Blanco, Verónica Pulito-Cueto, Raquel López-Mejías

**Affiliations:** 1Immunopathology Group, Rheumatology Department, Hospital Universitario Marqués de Valdecilla-IDIVAL, 39011 Santander, Spain; joabatis1995@gmail.com (J.C.B.-L.); fiorelfa@hotmail.com (V.C.-R.); msebastian@idival.org (M.S.M.-G.); ricardo.blanco@scsalud.es (R.B.); 2Division of Paediatrics, Hospital Universitario San Cecilio, 18016 Granada, Spain; belensev@hotmail.com; 3Instituto de Parasitología y Biomedicina ‘López-Neyra’, CSIC, PTS Granada, 18016 Granada, Spain; anamaort@ipb.csic.es (A.M.); javiermartin@ipb.csic.es (J.M.); 4Division of Paediatrics, Hospital Universitario Marqués de Valdecilla, 39008 Santander, Spain; maiteleonardo@hotmail.com (M.T.L.); anitapenalba@hotmail.com (A.P.); 5Departamento de Genética e Instituto de Biotecnología, Centro de Investigación Biomédica (CIBM), Universidad de Granada, 18071 Granada, Spain; dcarmona@ugr.es; 6Instituto de Investigación Biosanitaria ibs. Granada, 18012 Granada, Spain; 7Division of Rheumatology, Hospital Universitario de Bellvitge, 08907 Barcelona, Spain; fjnarvaez@bellvitgehospital.cat; 8Immunopathology Group, Division of Nephrology, Hospital Universitario Marqués de Valdecilla-IDIVAL, 39011 Santander, Spain; luismartinpenagos@gmail.com (L.M.-P.); belmarvega@outlook.es (L.B.-V.); 9Division of Dermatology, Hospital Universitario Marqués de Valdecilla, 39008 Santander, Spain; cristina.gomezf@scsalud.es; 10Internal Medicine Department, Hospital Universitario Central de Asturias, Instituto de Investigación Sanitaria del Principado de Asturias (ISPA), 33011 Oviedo, Spain; lcaminal@yahoo.es; 11Division of Rheumatology, Hospital Universitario Severo Ochoa, 28911 Madrid, Spain; paxko10@gmail.com; 12Division of Rheumatology, Hospital Universitario de La Princesa, IIS-Princesa, 28006 Madrid, Spain; pquiroga@alumni.unav.es (P.Q.-C.); miren_uriarte@hotmail.com (M.U.-E.); scastas@gmail.com (S.C.); 13Department of Rheumatology, Hospital Universitario Virgen del Rocío, 41013 Sevilla, Spain; cybereste@hotmail.com (E.R.); manuelleonluque@gmail.com (M.L.L.); 14Division of Rheumatology, Hospital Universitario de Basurto, 48013 Bilbao, Spain; juanmaria.blancomadrigal@osakidetza.net (J.M.B.-M.); evagalindez@gmail.com (E.G.-A.); 15Department of Rheumatology, IIS-Fundación Jiménez Díaz, 28040 Madrid, Spain; miguelaggay@hotmail.com; 16School of Medicine, Universidad de Cantabria, 39011 Santander, Spain

**Keywords:** IgA vasculitis, mucosal immune defence, polymorphisms

## Abstract

*ITGAM–ITGAX* (rs11150612, rs11574637), *VAV3* rs17019602, *CARD9* rs4077515, *DEFA* (rs2738048, rs10086568), and *HORMAD2* rs2412971 are mucosal immune defence polymorphisms, that have an impact on IgA production, described as risk *loci* for IgA nephropathy (IgAN). Since IgAN and Immunoglobulin-A vasculitis (IgAV) share molecular mechanisms, with the aberrant deposit of IgA1 being the main pathophysiologic feature of both entities, we assessed the potential influence of the seven abovementioned polymorphisms on IgAV pathogenesis. These seven variants were genotyped in 381 Caucasian IgAV patients and 997 matched healthy controls. No statistically significant differences were observed in the genotype and allele frequencies of these seven polymorphisms when the whole cohort of IgAV patients and those with nephritis were compared to controls. Similar genotype and allele frequencies of all polymorphisms were disclosed when IgAV patients were stratified according to the age at disease onset or the presence/absence of gastrointestinal or renal manifestations. Likewise, no *ITGAM–ITGAX* and *DEFA* haplotype differences were observed when the whole cohort of IgAV patients, along with those with nephritis and controls, as well as IgAV patients, stratified according to the abovementioned clinical characteristics, were compared. Our results suggest that mucosal immune defence polymorphisms do not represent novel genetic risk factors for IgAV pathogenesis.

## 1. Introduction

The mucosal immune system encompasses a series of defence mechanisms that protect the organism against infections [1,2]. Among them, immunoglobulin A (IgA) is described as the predominant immunoglobulin in the mucosa [1,2,3] with a crucial role in the humoral immune response [4], defending against microbial antigens [5]. Additionally, some pieces of evidence support the claim that genes also play a relevant role in the regulation of the mucosal immune response [2]. Accordingly, *ITGAM–ITGAX* rs11150612, *ITGAM–ITGAX* rs11574637, *VAV3* rs17019602, *CARD9* rs4077515, *DEFA* rs2738048, *DEFA* rs10086568, and *HORMAD2* rs2412971 are described as mucosal immune defence polymorphisms, most of which exhibit a relevant impact on IgA production by plasma cells in the mucosa [6,7,8,9,10]. 

The defining pathophysiologic feature of IgA vasculitis (IgAV), the most common small-sized blood vasculitis in children [11,12,13,14,15,16], is the presence of an aberrantly glycosylated galactose-deficient IgA (gd-IgA1) in circulation [17,18]. The origin of this gd-IgA1 is still unknown, although the intestinal mucosa seems to be essential in this process [19]. In this regard, IgA nephropathy (IgAN), the most prevalent primary chronic glomerular disease worldwide [20], is also characterised by an increased synthesis of gd-IgA1 [18,21,22]. In both entities, these elevated gd-IgA1 serum levels lead to glycan-specific IgG antibody development, which forms circulating immune complexes that ultimately deposit in different tissues, causing inflammation [23]. This evidence supports the hypothesis that IgAV and IgAN are inflammatory conditions that share pathophysiologic mechanisms. Interestingly, the seven mucosal immune defence genetic variants mentioned above are also proposed as risk *loci* for IgAN [6].

Taking all these considerations into account, it is plausible to think that polymorphisms, located in genes affecting the mucosal immune defence and reported as risk *loci* of IgAN [6], may be also implicated in the pathogenesis of IgAV. Accordingly, the main aim of the present study was to determine the potential influence of *ITGAM–ITGAX* rs11150612, *ITGAM–ITGAX* rs11574637, *VAV3* rs17019602, *CARD9* rs4077515, *DEFA* rs2738048, *DEFA* rs10086568, and *HORMAD2* rs2412971 genetic variants on the susceptibility and severity of IgAV, using the largest series of Caucasian patients diagnosed with IgAV ever assessed for genetic studies.

## 2. Results

The genotyping success rate was greater than 99% for the seven polymorphisms analysed.

No deviation from Hardy–Weinberg equilibrium (HWE) was detected for *ITGAM–ITGAX* (rs11150612, rs11574637), *VAV3* rs17019602, *CARD9* rs4077515, *DEFA* (rs2738048, rs10086568), and *HORMAD2* rs2412971 polymorphisms at the 5% significance level. 

Genotype and allele frequencies of all the genetic variants evaluated in our study were in accordance with those reported in the 1000 Genomes Project (http://www.internationalgenome.org/) (accessed on 11 August 2023) for European populations.

The linkage disequilibrium (LD) of *ITGAM–ITGAX* polymorphisms in our patients with IgAV (Figure 1A) and healthy controls (Figure 1B) was assessed. Moreover, the LD of *DEFA* polymorphisms in our study populations was also evaluated in our patients with IgAV (Figure 1A) and healthy controls (Figure 1B).

### 2.1. Mucosal Immune Defence Polymorphisms in the Susceptibility of IgAV

Firstly, we compared the genotype and allele frequencies of the seven polymorphisms evaluated as well as *ITGAM–ITGAX* and *DEFA* haplotype frequencies between patients with IgAV and healthy controls. 

In this context, no differences in the genotype and allele frequencies of *ITGAM–ITGAX* (rs11150612, rs11574637), *VAV3* rs17019602, *CARD9* rs4077515, *DEFA* (rs2738048, rs10086568), and *HORMAD2* rs2412971 were observed in patients with IgAV when compared to healthy controls (Table 1).

Moreover, similar *ITGAM–ITGAX* and *DEFA* haplotype frequencies were disclosed between patients with IgAV and healthy controls (Table 2).

In a further step, we analysed potential genetic differences in the seven polymorphisms assessed in those patients with IgAV who developed nephritis (IgAVN) compared to healthy controls. Accordingly, genotype and allele frequencies of all these variants (Table 3) and haplotype frequencies of *ITGAM–ITGAX* and *DEFA* (Table 4) did not differ between IgAVN patients and controls. 

### 2.2. Mucosal Immune Defence Polymorphisms in the Severity of IgAV

We evaluated whether differences in the genotype and allele frequencies of the seven polymorphisms evaluated as well as in the *ITGAM–ITGAX* and *DEFA* haplotype frequencies could exist between patients with IgAV stratified according to specific clinical characteristics of the disease.

In this sense, we analysed potential differences in the *ITGAM–ITGAX* (rs11150612, rs11574637), *VAV3* rs17019602, *CARD9* rs4077515, *DEFA* (rs2738048, rs10086568), and *HORMAD2* rs2412971 genotype and allele frequencies between patients with IgAV stratified according to the age at disease onset. No statistically significant differences were disclosed when the seven polymorphisms selected were compared between children (age ≤ 20 years) and adults (age > 20 years) (Table 5).

In addition, we evaluated whether differences in genotype and allele frequencies of the seven polymorphisms analysed differed between patients with IgAV stratified according to the presence/absence of gastrointestinal (GI) or renal manifestations. Accordingly, no statistically significant differences in *ITGAM–ITGAX* (rs11150612, rs11574637), *VAV3* rs17019602, *CARD9* rs4077515, *DEFA* (rs2738048, rs10086568), and *HORMAD2* rs2412971 genotype and allele frequencies were disclosed when patients with IgAV who developed GI manifestations were compared to those who did not exhibit these complications (Table 5). In addition, similar frequencies were disclosed when patients with IgAV were stratified according to the presence/absence of renal manifestations (Table 5).

The haplotype analyses did not yield additional information, since haplotype frequencies of *ITGAM–ITGAX* and *DEFA* did not differ between patients with IgAV stratified according to the age at the disease onset or the presence/absence of GI or renal manifestations (Table 6).

## 3. Discussion

This is the first study aimed to determine whether mucosal immune defence polymorphisms represent novel genetic risk factors for IgAV pathogenesis. To that aim, *ITGAM–ITGAX* (rs11150612, rs11574637), *VAV3* rs17019602, *CARD9* rs4077515, *DEFA* (rs2738048, rs10086568), and *HORMAD2* rs2412971 variants were evaluated, for the first time, in the largest series of Caucasian IgAV patients ever assessed for genetic studies. Given that these genetic variants have been previously reported as susceptibility *loci* of IgAN (4), and some of these exhibit functional consequences such as *ITGAM–ITGAX* rs11150612, that are related to increased expression of *ITGAX* in peripheral blood cells [24], and *CARD9* rs407751, associated with a higher expression of *CARD9* in different type of cells [24,25,26], it is plausible to think that may be also implicated in IgAV. 

Interestingly, our data revealed no influence of the seven polymorphisms mentioned on the susceptibility of IgAV, either when we studied each of these variants separately or when *ITGAM–ITGAX* and *DEFA* polymorphisms were tested together, conforming haplotypes. Moreover, no significant differences were disclosed when our patients with IgAV who developed renal damage were compared with healthy controls. These results are of great interest since no previous studies have evaluated the potential role of the seven abovementioned mucosal immune defence gene polymorphisms in the susceptibility of IgAV. Nevertheless, these polymorphisms have been described as susceptibility *loci* of IgAN, an inflammatory condition that shares pathogenic mechanisms with IgAV [27], although differences between them were also described [27]. Thus, whether these 2 conditions are different pathologies or represent different outcomes of a single disease is controversial [27]. In this sense, our findings shed light on this concern, showing no association of susceptibility IgAN *loci* with IgAV pathogenesis, mainly with the renal damage characteristic of this vasculitis, reinforcing the hypothesis that IgAV and IgAN may represent different entities. Indeed, we did not observe differences between patients with IgAV with nephritis and those patients with IgAV without renal manifestations. 

Furthermore, no statistically significant results were disclosed when genotype and allele frequencies of these seven polymorphisms and when *ITGAM–ITGAX* and *DEFA* haplotype frequencies were assessed according to the age at disease onset and the increased risk of GI disease or nephritis, suggesting that none of these genetic variants contributes to the severity of IgAV. It is important to note that no previous studies assessing the implication of the seven polymorphisms in the severity of this vasculitis are available, highlighting the relevance of our data in this regard.

In keeping with these findings, a previous study of our group evaluated the potential role of *TNFSF13* (another mucosal immune defence gene also described as a risk *locus* for IgAN (4)) and 2 *TNFSF13*-related genes in the pathogenesis of IgAV [28]. Results derived from this study showed a lack of implication of *TNFSF13* and *TNFSF13*-related genes with the susceptibility and severity of IgAV [28], highlighting that these genes do not contribute to the genetic network underlying IgAV.

Most mucosal immune defence genes encode proteins implicated in the maintenance of the intestinal barrier and regulation of mucosal immune response to pathogens [6]. *ITGAM* encodes integrins that mark intestinal dendritic cells [6] and is involved in the regulation of IgA-producing plasma cells [29]. VAV are crucial proteins implicated in NF-κB activation in B-cells [6], a process that stimulates IgA production [30], also required for proper differentiation of colonic enterocytes [31]. *CARD9* is essential for NF-κB activation, mediates intestinal repair, T-helper 17 responses, and control of bacterial infection after intestinal epithelial injury [32]. Mutations in *CARD9* are associated with ulcerative colitis, Crohn’s disease, and inflammatory bowel disease [33,34,35,36,37], diseases in which the integrative loss of the intestinal mucosal play a pathogenic role [38]. *DEFA* encodes α-defensins, antimicrobial peptides with a key role in mucosal defence [6], and is associated with Crohn’s disease [39,40]. Finally, *HORMAD2* is described as a protective *locus* against Crohn’s disease [33,34,41] and is associated with increased serum IgA levels [7]. Accordingly, and based on the results derived from our study, it is plausible that the genes affecting the mucosal immune defence evaluated in our work may not be the central defect in the pathogenesis of IgAV. In this line and given the pivotal role of galactosylation of IgA and the involvement of the complement pathway in the pathogenesis of IgAV, we hypothesize that polymorphisms in genes associated with these processes, also described as relevant loci involved in the pathogenesis of IgAN [42,43], may serve as better identifiers of IgAV. Since this vasculitis is a complex disease in which many genes may be implicated [44], future studies focused on elucidating the role of other molecular mechanisms in IgAV would be of great interest. 

## 4. Materials and Methods

### 4.1. Study Groups

A series of 381 unrelated Caucasian patients diagnosed with IgAV were enrolled in this study. All these patients fulfilled both Michel et al. [45] and the American College of Rheumatology [46] classification criteria for IgAV. Additionally, all the IgAV patients were recruited from Hospital Universitario Marqués de Valdecilla (Santander), Hospital Universitario Clínico San Cecilio (Granada), Hospital Universitario de Bellvitge (Barcelona), Hospital Universitario Lucus Augusti (Lugo), Hospital Universitario Central de Asturias (Oviedo), Hospital Universitario Severo Ochoa and Hospital Universitario de La Princesa (Madrid), Hospital Universitario Virgen del Rocío (Sevilla), and Hospital Universitario de Basurto (Bilbao). The clinical and demographical features of these patients were previously described [47].

Moreover, 997 unrelated individuals without a history of cutaneous vasculitis or any other autoimmune disease were enrolled in this work as healthy controls. All these individuals were sex- and ethnically-matched with IgAV patients, and their recruitment was performed in the National DNA Bank Repository (Salamanca). 

Informed written consent was obtained from all the patients diagnosed with IgAV and the healthy controls who were included in this study. Likewise, methods were carried out in accordance with the ethical standards of the approved guidelines and regulations, according to the Declaration of Helsinki, and the study was also approved by the Institutional Review Board (or Ethics Committee) of clinical research of Cantabria, Spain (protocol code 15/2012 and date of approval 11 May 2012).

### 4.2. Genotyping Method

Genomic deoxyribonucleic acid from all the patients with IgAV and healthy controls was extracted from peripheral blood samples using standard procedures. 

All individuals were genotyped for the following seven mucosal immune defence polymorphisms also described as risk *loci* for IgAN [6]: *ITGAM–ITGAX* rs11150612, *ITGAM–ITGAX* rs11574637, *VAV3* rs17019602, *CARD9* rs4077515, *DEFA* rs2738048, *DEFA* rs10086568, and *HORMAD2* rs2412971. Genotyping was performed using predesigned TaqMan 5′ single-nucleotide polymorphism genotyping assays (C_39031_20 for *ITGAM–ITGAX* rs11150612, C_30991393_10 for *ITGAM–ITGAX* rs11574637, C_34249468_20 for *VAV3* rs17019602, C_25956930_20 for *CARD9* rs4077515, C_27186146_10 for *DEFA* rs2738048, C_30155584_20 for *DEFA* rs10086568, and C_15795751_10 for *HORMAD2* rs2412971) in a QuantStudioTM 7 Flex Real-Time polymerase chain reaction system, according to the conditions recommended by the manufacturer (Applied Biosystems, Foster City, CA, USA).

To check the accuracy of the genotyping method, both negative controls and duplicate samples were evaluated. In addition, the genotyping success rate for all the genetic variants included in this study was tested, and the deviation of genotype data for the seven polymorphisms assessed from HWE was checked.

### 4.3. Statistical Analyses

Genotype and allele frequencies of *ITGAM–ITGAX* (rs11150612, rs11574637), *VAV3* rs17019602, *CARD9* rs4077515, *DEFA* (rs2738048, rs10086568), and *HORMAD2* rs2412971 were calculated and compared between patients with IgAV and healthy controls as well as between patients with IgAV stratified according to the specific clinical characteristic of the disease (age at the disease onset or presence/absence of GI or renal manifestation). For that analysis, a chi-squared test or Fisher test (when expected values were below 5) was used. The strength of association was estimated using odds ratio (OR) and 95% confidence intervals (CI).

Additionally, allelic combination (haplotype) analysis for the *ITGAM–ITGAX* and the *DEFA* polymorphisms evaluated were carried out. Haplotype frequencies were calculated using the Haploview v4.2 software (http://broad.mit.edu/mpg/haploview) (accessed on 11 August 2023) and compared between the groups mentioned above by chi-squared test. The strength of association was estimated by OR and 95% CI. *p*-values lower than 0.05 were considered statistically significant.

STATA statistical software 12/SE (Stata Corp., College Station, TX, USA) was used to perform all the statistical analyses.

## 5. Conclusions

In summary, our results, based on a large series of patients, suggest that mucosal immune defence polymorphisms do not represent novel genetic risk factors for the pathogenesis of IgAV.

## Figures and Tables

**Figure 1 ijms-24-13063-f001:**
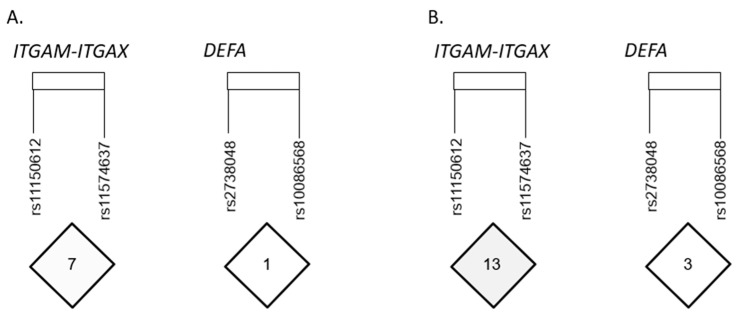
Linkage disequilibrium (LD) plot of *ITGAM–ITGAX* and *DEFA* polymorphisms in our patients with IgAV (**A**) and healthy controls (**B**) measured by r^2^ coefficient. Data were obtained using Haploview (v.4.2) software. The LD between the *ITGAM–ITGAX* and *DEFA* polymorphisms studied is shown on a scale from minimum (white) to maximum (black).

**Table 1 ijms-24-13063-t001:** Genotype and allele analysis of mucosal immune defence polymorphisms in patients with IgAV and healthy controls.

*Locus*	SNP	IgAV, % (n)	Healthy Controls, % (n)	*p*	OR [95% CI]	*Locus*	SNP	IgAV, % (n)	Healthy Controls, % (n)	*p*	OR [95% CI]
*ITGAM–ITGAX*	rs11150612					*DEFA*	rs2738048				
	GG	39.9 (152)	40.4 (403)	-	Ref.		AA	46.7 (178)	51.9 (517)	-	Ref.
	GA	43.3 (165)	46.3 (461)	0.69	0.95 [0.73–1.24]		AG	42.8 (163)	39.7 (396)	0.16	1.20 [0.92–1.55]
	AA	16.8 (64)	13.3 (133)	0.17	1.28 [0.88–1.84]		GG	10.5 (40)	8.4 (84)	0.12	1.38 [0.89–2.12]
	G	61.5 (469)	63.5 (1267)	-	Ref.		A	68.1 (519)	71.7 (1430)	-	Ref.
	A	38.5 (293)	36.5 (727)	0.94	1.09 [0.92–1.29]		G	31.9 (243)	28.3 (564)	0.07	1.19 [0.99–1.42]
*ITGAM–ITGAX*	rs11574637					*DEFA*	rs10086568				
	TT	69.0 (263)	65.3 (651)	-	Ref.		GG	43.6 (166)	42.3 (422)	-	Ref.
	TC	26.5 (101)	31.5 (314)	0.09	0.80 [0.60–1.05]		GA	44.9 (171)	45.8 (456)	0.71	0.95 [0.74–1.24]
	CC	4.5 (17)	3.2 (32)	0.37	1.31 [0.67–2.49]		AA	11.5 (44)	11.9 (119)	0.76	0.94 [0.62–1.41]
	T	82.3 (627)	81.0 (1616)	-	Ref.		G	66.0 (503)	65.2 (1300)	-	Ref.
	C	17.7 (135)	19.0 (378)	0.56	0.92 [0.74–1.14]		A	34.0 (259)	34.8 (694)	0.69	0.96 [0.81–1.15]
*VAV3*	rs17019602					*HORMAD2*	rs2412971				
	AA	61.7 (235)	59.7 (595)	-	Ref.		GG	29.4 (112)	27.3 (272)	-	Ref.
	AG	34.4 (131)	35.5 (354)	0.61	0.94 [0.72–1.21]		GA	52.0 (198)	51.2 (511)	0.66	0.94 [0.71–1.25]
	GG	3.9 (15)	4.8 (48)	0.44	0.79 [0.40–1.47]		AA	18.6 (71)	21.5 (214)	0.22	0.81 [0.56–1.16]
	A	78.9 (601)	77.4 (1544)	-	Ref.		G	55.4 (422)	52.9 (1055)	-	Ref.
	G	21.1 (161)	22.6 (450)	0.42	0.92 [0.75–1.13]		A	44.6 (340)	47.1 (939)	0.24	0.91 [0.77–1.07]
*CARD9*	rs4077515										
	CC	35.2 (134)	38.0 (379)	-	Ref.						
	CT	49.1 (187)	46.2 (461)	0.30	1.15 [0.88–1.50]						
	TT	15.7 (60)	15.8 (157)	0.67	1.08 [0.74–1.56]						
	C	59.7 (455)	61.1 (1219)	-	Ref.						
	T	40.3 (307)	38.9 (775)	0.49	1.06 [0.89–1.26]						

IgAV: IgA vasculitis; SNP: single nucleotide polymorphism; OR: Odds Ratio; CI: confidence interval.

**Table 2 ijms-24-13063-t002:** Haplotype analysis of mucosal immune defence polymorphisms in patients with IgAV and healthy controls.

*ITGAM–ITGAX*		IgAV, %	Healthy Controls, %	*p*	OR [95% CI]
rs11150612	rs11574637				
G	T	45.5	44.5	-	Ref.
A	T	36.8	36.5	0.89	0.99 [0.82–1.19]
G	C	16.0	19.0	0.12	0.83 [0.65–1.06]
*DEFA*		IgAV, % (n)	Healthy controls, % (n)	*p*	
rs2738048	rs10086568				
A	G	42.7	42.9	-	Ref.
A	A	25.4	28.8	0.26	0.89 [0.72–1.09]
G	G	23.3	22.3	0.64	1.05 [0.84–1.31]

IgAV: IgA vasculitis; OR: odds ratio; CI: confidence interval. Haplotypes with a frequency higher than 10% are displayed in the table.

**Table 3 ijms-24-13063-t003:** Genotype and allele analysis of mucosal immune defence polymorphisms in patients with IgAVN and healthy controls.

*Locus*	SNP	IgAVN, % (n)	Healthy Controls, % (n)	*p*	OR [95% CI]	*Locus*	SNP	IgAVN, % (n)	Healthy Controls, % (n)	*p*	OR [95% CI]
*ITGAM–ITGAX*	rs11150612					*DEFA*	rs2738048				
	GG	38.2 (50)	40.4 (403)	-	Ref.		AA	48.9 (64)	51.9 (517)	-	Ref.
	GA	44.3 (58)	46.3 (461)	0.95	1.01 [0.67–1.55]		AG	42.0 (55)	39.7 (396)	0.56	1.12 [0.75–1.68]
	AA	17.5 (23)	13.3 (133)	0.22	1.39 [0.78–2.43]		GG	9.1 (12)	8.4 (84)	0.67	1.15 [0.54–2.27]
	G	60.3 (158)	63.5 (1267)	-	Ref.		A	69.8 (183)	71.7 (1430)	-	Ref.
	A	39.7 (104)	36.5 (727)	0.31	1.15 [0.87–1.50]		G	30.2 (79)	28.3 (564)	0.53	1.09 [0.82–1.46]
*ITGAM–ITGAX*	rs11574637					*DEFA*	rs10086568				
	TT	72.5 (95)	65.3 (651)	-	Ref.		GG	42.7 (56)	42.3 (422)	-	Ref.
	TC	23.7 (31)	31.5 (314)	0.07	0.68 [0.43–1.05]		GA	44.3 (58)	45.8 (456)	0.83	0.96 [0.64–1.44]
	CC	3.8 (5)	3.2 (32)	0.89	1.07 [0.32–2.86]		AA	13.0 (17)	11.9 (119)	0.80	1.08 [0.56–1.96]
	T	84.4 (221)	81.0 (1616)	-	Ref.		G	64.9 (170)	65.2 (1300)	-	Ref.
	C	15.6 (41)	19.0 (378)	0.20	0.79 [0.54–1.13]		A	35.1 (92)	34.8 (694)	0.92	1.01 [0.77–1.34]
*VAV3*	rs17019602					*HORMAD2*	rs2412971				
	AA	63.4 (83)	59.7 (595)	-	Ref.		GG	30.5 (40)	27.3 (272)	-	Ref.
	AG	34.3 (45)	35.5 (354)	0.64	0.91 [0.60–1.36]		GA	48.1 (63)	51.2 (511)	0.41	0.84 [0.54–1.32]
	GG	2.3 (3)	4.8 (48)	0.17	0.48 [0.09–1.44]		AA	21.4 (28)	21.5 (214)	0.66	0.89 [0.51–1.53]
	A	80.5 (211)	77.4 (1544)	-	Ref.		G	54.6 (143)	52.9 (1055)	-	Ref.
	G	19.5 (51)	22.6 (450)	0.26	0.83 [0.59–1.15]		A	45.4 (119)	47.1 (939)	0.61	0.93 [0.72–1.22]
*CARD9*	rs4077515										
	CC	35.9 (47)	38.0 (379)	-	Ref.						
	CT	43.5 (57)	46.2 (461)	0.99	0.99 [0.65–1.54]						
	TT	20.6 (27)	15.8 (157)	0.21	1.39 [0.80–2.37]						
	C	57.6 (151)	61.1 (1219)	-	Ref.						
	T	42.4 (111)	38.9 (775)	0.28	1.16 [0.88–1.51]						

IgAVN: IgA vasculitis with nephritis; SNP: single nucleotide polymorphism; OR: Odds Ratio; CI: confidence interval.

**Table 4 ijms-24-13063-t004:** Haplotype analysis of mucosal immune defence polymorphisms in patients with IgAVN and healthy controls.

*ITGAM–ITGAX*		IgAVN, %	Healthy Controls, %	*p*	OR [95% CI]
rs11150612	rs11574637				
G	T	45.9	44.5	-	Ref.
A	T	38.4	36.5	0.84	1.03 [0.77–1.38]
G	C	14.4	19.0	0.13	0.74 [0.49–1.10]
*DEFA*		IgAVN, % (n)	Healthy Controls, % (n)	*p*	
rs2738048	rs10086568				
A	G	44.4	42.9	-	Ref.
A	A	25.5	28.8	0.35	0.86 [0.61–1.19]
G	G	20.5	22.3	0.52	0.89 [0.62–1.27]

IgAVN: IgA vasculitis with nephritis; OR: odds ratio; CI: confidence interval. Haplotypes with a frequency higher than 10% are displayed in the table.

**Table 5 ijms-24-13063-t005:** Genotype and allele analysis of mucosal immune defence polymorphisms in patients with IgAV stratified according to clinical characteristics.

		Age of Onset				GI Manifestations ^1^				Renal Manifestations ^2^			
*Locus*	SNP	Children, % (n)	Adults, % (n)	*p*	OR [95% CI]	Yes, % (n)	No, % (n)	*p*	OR [95% CI]	Yes, % (n)	No, % (n)	*p*	OR [95% CI]
*ITGAM–ITGAX*	rs11150612												
	GG	39.1 (108)	41.9 (44)	-	Ref.	43.4 (85)	35.9 (66)	-	Ref.	38.2 (50)	40.8 (102)	-	Ref.
	GA	42.4 (117)	45.7 (48)	0.98	0.99 [0.59–1.66]	41.8 (82)	45.1 (83)	0.24	0.77 [0.48–1.22]	44.3 (58)	42.8 (107)	0.67	1.10 [0.68–1.81]
	AA	18.5 (51)	12.4 (13)	0.19	1.60 [0.76–3.21]	14.8 (29)	19.0 (35)	0.14	0.64 [0.34–1.21]	17.5 (23)	16.4 (41)	0.67	1.14 [0.59–2.20]
	G	60.3 (333)	64.8 (136)	-	Ref.	64.3 (252)	58.4 (215)	-	Ref.	60.3 (158)	62.2 (311)	-	Ref.
	A	39.7 (219)	35.2 (74)	0.26	1.21 [0.86–1.71]	35.7 (140)	41.6 (153)	0.10	0.78 [0.58–1.06]	39.7 (104)	37.8 (189)	0.61	1.08 [0.79–1.49]
*ITGAM–ITGAX*	rs11574637												
	TT	68.1 (188)	71.4 (75)	-	Ref.	65.8 (129)	72.3 (133)	-	Ref.	72.5 (95)	67.2 (168)	-	Ref.
	TC	27.6 (76)	23.8 (25)	0.47	1.21 [0.70–2.15]	29.1 (57)	23.9 (44)	0.22	1.34 [0.82–2.18]	23.7 (31)	28.0 (70)	0.33	0.78 [0.46–1.31]
	CC	4.3 (12)	4.8 (5)	0.94	0.96 [0.30–3.59]	5.1 (10)	3.8 (7)	0.44	1.47 [0.49–4.70]	3.8 (5)	4.8 (12)	0.58	0.74 [0.20–2.34]
	T	81.9 (452)	83.3 (175)	-	Ref.	80.4 (315)	84.2 (310)	-	Ref.	84.4 (221)	81.2 (406)	-	Ref.
	C	18.1 (100)	16.7 (35)	0.64	1.11 [0.71–1.74]	19.6 (77)	15.8 (58)	0.16	1.31 [0.88–1.94]	15.6 (41)	18.8 (94)	0.28	0.80 [0.52–1.22]
*VAV3*	rs17019602												
	AA	60.5 (167)	64.8 (68)	-	Ref.	60.7 (119)	63.0 (116)	-	Ref.	63.4 (83)	60.8 (152)	-	Ref.
	AG	35.5 (98)	31.4 (33)	0.44	1.21 [0.73–2.03]	34.7 (68)	33.7 (62)	0.76	1.07 [0.68–1.68]	34.3 (45)	34.4 (86)	0.85	0.96 [0.60–1.54]
	GG	4.0 (11)	3.8 (4)	0.85	1.12 [0.32–4.98]	4.6 (9)	3.3 (6)	0.48	1.46 [0.45–5.15]	2.3 (3)	4.8 (12)	0.23	0.46 [0.08–1.77]
	A	78.3 (432)	80.5 (169)	-	Ref.	78.1 (306)	79.9 (294)	-	Ref.	80.5 (211)	78.0 (390)	-	Ref.
	G	21.7 (120)	19.5 (41)	0.50	1.14 [0.76–1.75]	21.9 (86)	20.1 (74)	0.54	1.12 [0.78–1.61]	19.5 (51)	22.0 (110)	0.42	0.86 [0.58–1.26]
*CARD9*	rs4077515												
	CC	37.0 (102)	30.5 (32)	-	Ref.	37.8 (74)	32.6 (60)	-	Ref.	35.9 (47)	34.8 (87)	-	Ref.
	CT	47.1 (130)	54.3 (57)	0.19	0.72 [0.42–1.22]	42.3 (83)	56.0 (103)	0.06	0.65 [0.41–1.05]	43.5 (57)	52.0 (130)	0.39	0.81 [0.49–1.34]
	TT	15.9 (44)	15.2 (16)	0.68	0.86 [0.41–1.87]	19.9 (39)	11.4 (21)	0.20	1.51 [0.77–3.00]	20.6 (27)	13.2 (33)	0.19	1.51 [0.77–2.94]
	C	60.5 (334)	57.6 (121)	-	Ref.	58.9 (231)	60.6 (223)	-	Ref.	57.6 (151)	60.8 (304)	-	Ref.
	T	39.5 (218)	42.4 (89)	0.47	0.89 [0.63–1.24]	41.1 (161)	39.4 (145)	0.64	1.07 [0.79–1.45]	42.4 (111)	39.2 (196)	0.40	1.14 [0.83–1.56]
*DEFA*	rs2738048												
	AA	44.9 (124)	51.4 (54)	-	Ref.	46.4 (91)	47.3 (87)	-	Ref.	48.9 (64)	45.6 (114)	-	Ref.
	AG	43.5 (120)	41.0 (43)	0.42	1.22 [0.74–2.01]	43.9 (86)	41.3 (76)	0.72	1.08 [0.69–1.69]	42.0 (55)	43.2 (108)	0.67	0.91 [0.57–1.45]
	GG	11.6 (32)	7.6 (8)	0.19	1.74 [0.72–4.66]	9.7 (19)	11.4 (21)	0.68	0.86 [0.41–1.82]	9.1 (12)	11.2 (28)	0.48	0.76 [0.33–1.68]
	A	66.7 (368)	71.9 (151)	-	Ref.	68.4 (268)	67.9 (250)	-	Ref.	69.8 (183)	67.2 (336)	-	Ref.
	G	33.3 (184)	28.1 (59)	0.17	1.28 [0.89–1.85]	31.6 (124)	32.1 (118)	0.90	0.98 [0.71–1.35]	30.2 (79)	32.8 (164)	0.46	0.88 [0.63–1.24]
*DEFA*	rs10086568												
	GG	43.5 (120)	43.8 (46)	-	Ref.	45.4 (89)	41.3 (76)	-	Ref.	42.7 (56)	44.0 (110)	-	Ref.
	GA	47.1 (130)	39.1 (41)	0.43	1.22 [0.72–2.04]	43.9 (86)	46.2 (85)	0.50	0.86 [0.55–1.36]	44.3 (58)	45.2 (113)	0.97	1.01 [0.63–1.62]
	AA	9.4 (26)	17.1 (18)	0.09	0.55 [0.26–1.18]	10.7 (21)	12.5 (23)	0.46	0.78 [0.38–1.60]	13.0 (17)	10.8 (27)	0.54	1.24 [0.58–2.58]
	G	67.0 (370)	63.3 (133)	-	Ref.	67.3 (264)	64.4 (237)	-	Ref.	64.9 (170)	66.6 (333)	-	Ref.
	A	33.0 (182)	36.7 (77)	0.34	0.85 [0.60–1.20]	32.7 (128)	35.6 (131)	0.39	0.88 [0.64–1.20]	35.1 (92)	33.4 (167)	0.64	1.08 [0.78–1.50]
*HORMAD2*	rs2412971												
	GG	27.5 (76)	34.3 (36)	-	Ref.	29.6 (58)	29.3 (54)	-	Ref.	30.5 (40)	28.8 (72)	-	Ref.
	GA	52.9 (146)	49.5 (52)	0.27	1.33 [0.77–2.27]	51.5 (101)	52.2 (96)	0.93	0.98 [0.60–1.60]	48.1 (63)	54.0 (135)	0.48	0.84 [0.50–1.41]
	AA	19.6 (54)	16.2 (17)	0.23	1.50 [0.73–3.16]	18.9 (37)	18.5 (34)	0.97	1.01 [0.54–1.92]	21.4 (28)	17.2 (43)	0.61	1.17 [0.60–2.26]
	G	54.0 (298)	59.0 (124)	-	Ref.	55.4 (217)	55.4 (204)	-	Ref.	54.6 (143)	55.8 (279)	-	Ref.
	A	46.0 (254)	41.0 (86)	0.21	1.23 [0.88–1.72]	44.6 (175)	44.6 (164)	0.98	1.00 [0.75–1.35]	45.4 (119)	44.2 (221)	0.75	1.05 [0.77–1.43]

IgAV: IgA vasculitis; SNP: single-nucleotide polymorphism; GI: gastrointestinal. ^1^ Bowel angina and/or gastrointestinal bleeding; ^2^ Hematuria, proteinuria, or nephrotic syndrome at any time over the clinical course of the disease and/or renal sequelae (persistent renal involvement) at the last follow-up.

**Table 6 ijms-24-13063-t006:** Haplotype analysis of mucosal immune defence polymorphisms in patients with IgAV stratified according to clinical characteristics.

		Age of Onset				GI Manifestations ^1^				Renal Manifestations ^2^			
*ITGAM–ITGAX*		Children, %	Adults, %	*p*	OR [95% CI]	Yes, %	No, %	*p*	OR [95% CI]	Yes, %	No, %	*p*	OR [95% CI]
rs11150612	rs11574637												
G	T	43.9	49.9	-	Ref.	45.9	44.7	-	Ref.	45.9	45.3	-	Ref.
A	T	38.0	33.4	0.14	1.30 [0.90–1.89]	34.4	39.5	0.32	0.85 [0.61–1.18]	38.4	35.9	0.72	1.06 [0.75–1.49]
G	C	16.5	14.8	0.31	1.27 [0.78–2.11]	18.4	13.7	0.19	1.32 [0.85–2.05]	14.4	16.9	0.49	0.86 [0.53–1.36]
*DEFA*		Children, % (n)	Adults, % (n)	*p*	OR [95% CI]	Yes, % (n)	No, % (n)	*p*	OR [95% CI]	Yes, % (n)	No, % (n)	*p*	OR [95% CI]
rs2738048	rs10086568												
A	G	43.2	41.4	-	Ref.	45.1	40.0	-	Ref.	44.4	41.9	-	Ref.
A	A	23.4	30.5	0.14	0.75 [0.50–1.12]	23.2	28.0	0.09	0.73 [0.51–1.07]	25.5	25.3	0.81	0.96 [0.65–1.41]
G	G	23.9	21.9	0.89	1.03 [0.67–1.60]	22.2	24.4	0.24	0.80 [0.55–1.18]	20.5	24.7	0.25	0.79 [0.53–1.20]

IgAV: IgA vasculitis; OR: odds ratio; CI: confidence interval; GI: Gastrointestinal. ^1^ Bowel angina and/or gastrointestinal bleeding; ^2^ Hematuria, proteinuria, or nephrotic syndrome at any time over the clinical course of the disease and/or renal sequelae (persistent renal involvement) at the last follow-up. Haplotypes with a frequency higher than 10% are displayed in the table.

## Data Availability

All data generated or analysed during this study are included in this published article.

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
