# Peer review of "Mucosal Immune Defence Gene Polymorphisms as Relevant Players in the Pathogenesis of IgA Vasculitis?"

_ijms, 2023, doi:10.3390/ijms241713063_

Round 1

Reviewer 1 Report

The Authors provided an interesting study about mucosal immune defence genetic polymorphism related to onset of IgA nephropathy (IgAN). Since IgAN and Immunoglobulin-A vasculitis (IgAV) share molecular mechanisms, being the aberrant deposit of IgA1 the main pathophysiologic feature of both entities, they aimed to assess the potential influence of the seven polymorphisms above-mentioned on IgAV pathogenesis. Although mucosal immune defence polymorphisms do not represent novel genetic risk factors for IgAV pathogenesis, the manuscript is well written and organized. 

Reviewer 2 Report

The study shows that intestinal bacterial specific immune response associated gene polymorphisms do not play a role in identifying IgAV patients as compared to controls. Although the study did not find an associated of IgAV with any of the allele distribution the manuscript can be made better by discussing about some of the published data using the same gene polymorphisms in a different cohort of IgAV patients.

With respect to this, ITGAM-ITGAX 64 rs11150612, ITGAM-ITGAX rs11574637, VAV3 rs17019602, CARD9 rs4077515, DEFA 65 rs2738048, DEFA rs10086568, and HORMAD2 rs2412971 are described as mucosal im-66 mune defence polymorphisms, exhibiting most of them a relevant impact on IgA produc-67 tion by plasma cells in the mucosa [6].

Provide more references.

 Mucosal immune defence polymorphisms as relevant players in 2 the pathogenesis of IgA vasculitis?

Adding gene before polymorphisms in title seems more apt

A discussion about polymorphism reported in genes responsible for IgA galactosylation (glycosyltransferase enzymes) and complement pathway associated gene polymorphism may serve as better identifiers of IgAV. For instance a study by Ibrahim et al group ‘Genetic polymorphism in C3 is associated with progression in chronic kidney disease (CKD) patients with IgA nephropathy but not in other causes of CKD’

Discuss about studies done in a different cohort of IgAV patients using same gene polymorphisms.

The quality of English language is fine.
